# IM-Context: In-Context Learning for Imbalanced Regression Tasks

**Ismail Nejjar**                                                                          *ismail.nejjar@epfl.ch*
*École Polytechnique Fédérale de Lausanne (EPFL) and Massachusetts Institute of Technology (MIT)*

**Faez Ahmed**                                                                                  *faez@mit.edu*
*Massachusetts Institute of Technology (MIT)*

**Olga Fink**                                                                                *olga.fink@epfl.ch*
*École Polytechnique Fédérale de Lausanne (EPFL)*

**Reviewed on OpenReview:** *https://openreview.net/forum?id=p4Y844vJWG*

## Abstract

Regression models often fail to generalize effectively in regions characterized by highly imbalanced label distributions. Previous methods for deep imbalanced regression rely on gradient-based weight updates, which tend to overfit in underrepresented regions. This paper proposes a paradigm shift towards in-context learning as an effective alternative to conventional in-weight learning methods, particularly for addressing imbalanced regression. In-context learning refers to the ability of a model to condition itself, given a prompt sequence composed of in-context samples (input-label pairs) alongside a new query input to generate predictions, without requiring any parameter updates. In this paper, we study the impact of the prompt sequence on the model performance from both theoretical and empirical perspectives. We emphasize the importance of localized context in reducing bias within regions of high imbalance. Empirical evaluations across a variety of real-world datasets demonstrate that in-context learning substantially outperforms existing in-weight learning methods in scenarios with high levels of imbalance.

## 1 Introduction

Imbalanced data distributions, common in the real world, pose significant challenges to the generalization of conventional deep learning models due to variance across minority labels and bias toward majority labels (Zhang et al., 2023). While numerous studies have addressed learning from imbalanced data, most of them focus on classification tasks (Yang & Xu, 2020; Wang et al., 2024; Zhang et al., 2022; Yang et al., 2024). Recent work emphasizes that the continuity of the label space, and the relationship between features and labels (Shin et al., 2022), make imbalanced regression a fundamentally different problem from imbalanced classification (Yang et al., 2021; Ren et al., 2022). Imbalanced regression problems are critical in many fields. For instance, in computer vision, age estimation datasets are often imbalanced, with fewer samples for younger age groups due to legal and ethical limitations, while older age groups are underrepresented as the population naturally declines with age. Similarly, in engineering design, the distribution of designs is often skewed, with the most desirable characteristics at the tail end of the distribution. This means that the region of greatest interest lies in the minority samples of the distribution. Enhancing model performance in these low-data regions is essential for achieving more accurate and reliable outcomes across these applications.

To address the specificities of deep imbalanced regression, two main approaches have been proposed: sample re-weighting and embedding space regularization. Sample re-weighting techniques, as those proposed by Yang et al. (2021) and Steininger et al. (2021), apply kernel density estimation to smooth the label distribution. This method leverages local dependencies by enforcing similarity between nearby labels. Alternatively, embedding

space regularization techniques, such as ranking (Gong et al., 2022) and contrastive learning (Keramati et al., 2024), preserve both local and global dependencies by leveraging label similarity rankings within the feature space. These methods, commonly referred to as **in-weight learning**, rely on gradient updates to adjust model weights, assuming that models can effectively generalize from limited data points in tail regions. However, with inherently less information in minority regions of the label distribution, these models are prone to overfitting samples from these regions (Cao et al., 2019).

A different learning paradigm, the recently proposed **in-context learning**, has the potential to overcome the limitations of **in-weight learning** in minority regions. In-context learning refers to the ability of a model to rapidly generalize to the new concepts on which it has not been previously trained (Reddy, 2023; Bietti et al., 2024), using only a few examples—often referred to as few-shot (Brown et al., 2020; Webb et al., 2023). This approach is particularly notable because it does not require any parameter updates on the model's weights.

Recent work has explored **in-context learning for regression tasks**. For instance, the recently proposed 'Prior-data Fitted Networks' (Müller et al., 2022) were trained to mimic Bayesian inference, similar to Gaussian processes and Bayesian neural networks, and applied to Bayesian optimization (Müller et al., 2023). Additionally, it has been empirically shown that transformers can learn from scratch using randomly sampled data from function classes, achieving performance comparable to that of the optimal least squares estimator in linear function scenarios (Garg et al., 2022). Moreover, transformers can even learn from more complex function classes (Garg et al., 2022). Interestingly, Chan et al. (2022) highlight that in-context learning particularly emerges in training distributions with properties such as long-tailedness. *Unlike **in-weight learning**, which needs to be trained from scratch on specific datasets, an **in-context learning** model is pre-trained on diverse, often synthetic, data and can adapt to different tasks using context examples. This means a single model can perform multiple tasks without needing a separate model for each one.*

In this paper, we introduce In-Context Learning for IMbalanced Regression Tasks, referred to as IM-Context, to explore this new learning paradigm from an imbalanced regression perspective. We present the counter-intuitive finding that increasing the context size can negatively impact imbalanced regression performance in minority regions. We establish a theoretical error bound for the expected error when in-context learning is applied to imbalanced regression, demonstrating that a large number of context examples can theoretically approximate the true predictive distribution. However, in the case of imbalanced regression, using the entire training set as context biases the model toward the majority region, leading to inferior performance on rare labels. As an alternative, we propose a localized approach where only the 'closest' in-context samples to a new query are used. This method mitigates bias and reduces the sequence length of context samples, thereby reducing the memory requirement for each inference.

In this localized setting, our findings reveal that in regions with dense label distributions (where the majority of samples are located), the expected error remains tightly bounded and relatively flat, indicating insensitivity to the number of context samples. Conversely, in regions with sparse label distributions (minority of samples), we observe a shift in behavior where more context samples hurt the performances in the minority region: the bound widens as more context samples are added. This highlights the need for locality, as indiscriminately adding more context examples can skew the context label distribution and worsen predictions. However, retrieving semantically relevant examples as context in practice is challenging, particularly for imbalanced datasets, because the probability of retrieving samples from the majority region is higher. To address this challenge, we propose creating a second training set, where the number of samples in each region is inversely proportional to its representation in the original set: majority regions have fewer points, while minority regions are overrepresented, we refer to it as inverse density dataset. Given a new query sample, we retrieve neighboring samples from both training sets. The proposed strategy, referred to as Augmented, serves to mitigate potential biases toward the majority region; N.B. alternative sampling strategies could be applied.

To empirically validate these findings, we use two pre-trained models (Garg et al., 2022; Müller et al., 2023), and evaluate our methodology across eight imbalanced regression tasks. This comprises three benchmark datasets covering two facial age estimation tasks, one text similarity task, and six tabular datasets with different degrees of imbalance. Experiments in several real-world datasets show that our in-context learning approach consistently outperforms state-of-the-art in-weight learning methods, particularly in regions with high imbalance. The code is available `https://github.com/ismailnejjar/IM-Context`.

## 2 Related works

**Imbalanced regression:** The continuity in label space makes imbalanced regression different from imbalanced classification. One potential option is to discretize the continuous label space and apply methods typically used in imbalanced classification. However, this requires specifying a minimum acceptable error threshold for the discretization process (Ren et al., 2022). Moreover, the continuity of the label space can provide additional information about the relationships between data instances, contrasting with the categorical distinctions in classification tasks (Yang et al., 2021).

Various strategies have been proposed to refine how relationships in both label and feature spaces are handled in regression. Label Distribution Smoothing (LDS) (Yang et al., 2021) and DenseLoss (Steininger et al., 2021) employed kernel density estimation to derive the 'true' label density from empirical data, thereby smoothing and reweighting the data labels. Feature Distribution Smoothing (FDS) (Yang et al., 2021) applied similar principles to the feature space and performs distribution smoothing by transferring the first two feature statistics between nearby target labels. A different direction to overcome deep imbalanced regression is to learn additional tasks to regularise the model feature representation. Different types of tasks have been proposed to capture the relationships between features and labels at local and global levels (Gong et al., 2022; Keramati et al., 2024; Wang & Wang, 2024). For instance, Gong et al. (2022) proposed ranking similarity regularization, while Keramati et al. (2024) adapted contrastive learning to regression tasks by improving the continuity of the feature space. In a recent study, Wang & Wang (2024) leveraged deep evidential regression (Amini et al., 2020)(a framework that learns to predict uncertainty in a single forward pass) and proposed leveraging data with similar target labels to compute the variational distribution of the latent representation, imposing probabilistic reweighting on imbalanced data.

However, these deep learning methods predominantly relied on **in-weight learning**, wherein achieving high-quality representations in a long-tailed setting is difficult because the features in the minority can be easily overfitted. This issue was highlighted by Ren et al. (2022), as the Mean Square Error (MSE) loss used for training in imbalanced regression introduces biases to the majority region. Alternative learning strategies may potentially help overcome the limitation of in-weight learning in minority regions.

**In-context Learning (ICL) in regression:** ICL, based on Transformer architecture, enables a pre-trained model to learn a new task with minimal examples (Vaswani et al., 2017; Brown et al., 2020). The remarkable success of transformers and their ability to do in-context learning shown in natural language processing has inspired a line of research exploring the algorithmic power of transformers. Notably, Garg et al. (2022) investigated transformers in in-context regression tasks, ranging from learning linear regression to more complex function classes. Subsequent works have suggested that the attention mechanism in the transformer may mimic various optimization algorithms such as gradient (Akyurek et al., 2023; Von Oswald et al., 2023; Ahn et al., 2024; Mahankali et al., 2024; Bai et al., 2023).

From a Bayesian perspective, Xie et al. (2022) suggested that in-context learning is a form of implicit Bayesian inference. Building on this idea, Müller et al. (2022) trained transformer models using 'Prior Fitted Networks', where data sampled from a prior distribution enables these models to approximate the posterior predictive distribution during inference for tabular data (van Breugel & van der Schaar, 2024). A theoretical foundation for these models was further developed by Nagler (2023) with a focus on classification tasks. However, the practical implementation of ICL in regression, particularly under conditions of extreme data imbalance or when dealing with real-world data, remains underexplored.

## 3 METHODOLOGY

### 3.1 Problem Setting

We address a regression problem where an input feature $\mathbf{x} \in \mathcal{X} \subseteq \mathbb{R}^d$ is mapped to a label $y \in \mathcal{Y} = \mathbb{R}$. More specifically, in the imbalanced regression scenario, the training set $D_s = \{(\mathbf{x}_i, y_i)\}_{i=1}^{n_s}$ and the test set $D_t = \{(\mathbf{x}_i, y_i)\}_{i=1}^{n_t}$ are sampled from distinct joint distributions, $p_s(\mathbf{x}, y)$ and $p_t(\mathbf{x}, y)$, referred to as the source and target distributions, respectively. In our setup, the source distribution typically exhibits a skewed label

distribution $p_s(y)$, while the target distribution is uniformly spread across the range of target values $p_t(y)$. The goal is to improve the prediction on the tail regions.

**In-Context Learning** Consider a transformer model $f_\theta$ which is capable of in-context learning. This capability is demonstrated when the model can accurately approximate the output $y_{query}$ for a new query input $\mathbf{x}_{query}$ based on a sequence of in-context examples. This sequence, called a prompt, is given by $(\mathbf{x}_1, y_1, \ldots, \mathbf{x}_n, y_n, \mathbf{x}_{query})$. The training set $D_s$ can serve as a basis for the prompt sequence. Given a new query sample $\mathbf{x}_{query}$ from the test set $D_t$, the model predicts $\hat{y}_{query} = f_\theta(\mathbf{x}_{query} \mid D_s)$. In the following sections, we evaluate the circumstances under which $\hat{y}_{query}$ can be accurately similar to the true label $y_{query}$.

## 3.2 Convergence

In the following sections, we delve into the theoretical analysis of the model convergence in the general case. We show that in imbalance scenarios, using the entire training set can introduce bias, primarily due to the overrepresentation of the majority samples. We introduce a localized version aimed at alleviating this bias and provide a theoretical bound for the expected prediction error, showing that the behavior differs depending on whether points are from majority or minority regions.

### 3.2.1 General convergence

Given a pre-trained transformer capable of in-context learning $f_\theta$, let us consider the case where we sample points $D_n$ from a realization of the random training distribution $D_s$. We can decompose the model's Expected Prediction Error (EPE) for a new sample $(\mathbf{x}, y)$ as :

$$
\begin{aligned}
\text{EPE}_{f_\theta}(x) &= \mathbb{E}_{D_n}\left[(f_\theta(\mathbf{x} \mid D_n) - \mathbb{E}[y \mid \mathbf{x}])^2\right] \\
&= \mathbb{E}_{D_n}[((f_\theta(\mathbf{x} \mid D_n) - \mathbb{E}_{D_n}[f_\theta(\mathbf{x} \mid D_n)]) + (\mathbb{E}_{D_n}[f_\theta(\mathbf{x} \mid D_n)] - \mathbb{E}[y \mid \mathbf{x}]))^2] \quad (1) \\
&= \text{Var}_{D_n}[f_\theta(\mathbf{x} \mid D_n)] + \left(\text{Bias}^2_{D_n}[f_\theta(\mathbf{x} \mid D_n)]\right) + \sigma^2
\end{aligned}
$$

**Proposition 3.1** *c-Lipschitz Continuity: Consider an infinitely large training set $D_s$, from which subsets $D_n$ and $D'_n$ are independently sampled. Assume that the label noise is constant. Given the observation from Garg et al. (2022) that error decreases as more samples are given as context, we can consider that the model $f_\theta$, is c-Lipschitz, with $c = (c_1, \ldots, c_n) \in \mathbb{R}^n_+$ where $c_i = \delta i^{-\alpha}$ for each $i \in \{1, \ldots, n\}$, with $\delta$ as a positive constant and $\alpha > 0.5$:*

$$
|f_\theta(\mathbf{x} \mid D_n) - f_\theta(\mathbf{x} \mid D'_n)| \leq \sum_{i=1}^{n} c_i \mathbf{1}_{\{\mathbf{x}_i \neq \mathbf{x}'_i\}} \quad (2)
$$

**Theorem 3.2** *Application of McDiarmid's Inequality McDiarmid et al. (1989) : Given that $f_\theta$ is c-Lipschitz under the defined conditions, for any $t > 0$, the tail probability is bounded by:*

$$
\Pr(|f_\theta(\mathbf{x} \mid D_n) - \mathbb{E}[f_\theta(\mathbf{x} \mid D_n)]| \geq t) \leq 2 \exp\left(-\frac{2t^2}{\|c\|_2^2}\right), \quad (3)
$$

Since $\|c\|_2^2 = \sum_{i=1}^{\infty}(\delta i^{-\alpha})^2$, which converges given $\alpha > 0.5$, by the Borel-Cantelli lemma, $f_\theta(x \mid D_n)$ convergences to the expected prediction almost surely:

$$
f_\theta(\mathbf{x} \mid D_n) \xrightarrow{\text{a.s.}} \mathbb{E}[f_\theta(\mathbf{x} \mid D_n)] \quad \text{as} \quad n \to \infty, \quad (4)
$$

**Variance :** Under the assumption of **Proposition 1**, the variance of the model's prediction should decrease as more samples are provided.

**Proposition 3.3** *Convergence to training label distribution: If we assume that the attention blocks in model $f_\theta$ are 'mimicking' gradient descent steps (Von Oswald et al., 2023; Ahn et al., 2024) on the context example $D_n$, the model's behavior should resemble that of a neural network trained with Mean Square Error loss. Consequently, we can expect:*

$$
\mathbb{E}_{D_n}[f_\theta(\mathbf{x} \mid D_n)] \to \mathbb{E}_{p_s}[y \mid \mathbf{x}] \quad \text{as} \quad n \to \infty \quad (5)
$$

**Bias :** While **Proposition 3.3** suggests that the model converges to the conditional probability of $p_s$, the behavior of $f_\theta$ crucially depends on $p_s$. If the samples from $D_n$ are independently and identically distributed, $f_\theta$ can be considered an unbiased estimator of $\mathbb{E}_{p_s}[y \mid \mathbf{x}]$. Conversely, if $D_n$ contains biased or non-representative samples, as often occurs in imbalanced regression scenarios, the model's predictions will inherently reflect these biases, and the model will tend to underestimate rare labels.

### 3.2.2 Local convergence

**Imbalanced regression :** A simple approach to reduce model bias in the tail regions of the label distribution is to select a finite set of neighboring samples of $\mathbf{x}$ from $D_n$. Selecting only the neighboring sample for context is equivalent to adjusting the distribution of labels $p_s(y \mid \mathbf{x})$ so that it corresponds locally to $p_t(y \mid \mathbf{x})$. This reflects the proportional relationship $\frac{p_s(y \mid \mathbf{x})}{p_t(y \mid \mathbf{x})} \approx \frac{p_s(y)}{p_t(y)}$.

While previous propositions ensure convergences for infinitely large datasets, it is also important to consider the expected prediction error in the case of small and finite numbers of context examples. Under this condition, we can assume that, in the worst case, the model will simply average the labels of the context samples. To the best of our knowledge, there is no existing work that specifically addresses this issue in the context of imbalanced regression with k-NN. This assumption is in line with previous works, for example Garg et al. (2022) showed that transformers always outperform sample averaging, while Bhattamishra et al. (2023) demonstrated that a frozen GPT-2 can mimic the nearest neighbor algorithm. Formally, with $k$ samples from $D_n$ closest to $\mathbf{x}$, denoted as $D_k$, the expected error bound is given by:

$$\mathbb{E}_{D_k}\left[(\mathbb{E}[y \mid \mathbf{x}] - f_\theta(\mathbf{x} \mid D_k))^2\right] \leq \mathbb{E}_{D_k}\left[(\mathbb{E}[y \mid \mathbf{x}] - \tilde{f}(\mathbf{x} \mid D_k))^2\right] \tag{6}$$

where $\tilde{f}(\mathbf{x} \mid D_k) = \frac{1}{k}\sum_{i=1}^{k} y_i$ represents the average of in-context labels, with $k > 1$. The associated error can be decomposed following Hastie et al. (2009) into:

$$\mathbb{E}_{D_k}\left[(\mathbb{E}[y \mid \mathbf{x}] - f_\theta(\mathbf{x} \mid D_k))^2\right] \leq \underbrace{\left(y - \frac{1}{k}\sum_{i=1}^{k} y_i\right)^2}_{\text{Bias}} + \underbrace{\frac{\sigma^2}{k}}_{\text{Variance}} + \sigma^2 \tag{7}$$

In the case of imbalanced regression, the variance term depends on the number of selected neighbors $k$ and should decrease as more context examples are selected, similar to before. The bias for the minority regions will increase with $k$ as the averaging estimator will predict the mean of the context examples, which is also in agreement with the previous propositions. Figure 1 illustrates this behavior by plotting the expected error for data points across different regions, for the Boston and AgeDB datasets. For points in the majority region, the expected error decreases as more context examples are given. **However, for points in the tail region, the error initially decreases but then exhibits a U-shaped curve as $k$ increases.** Thus, for minority samples, the error bound is tighter for a small number of selected neighbors, further emphasizing the need for localized context.

### 3.2.3 Empirical validation

To validate the error bound in practice, we compared the results obtained on two datasets: Boston and AgeDB, across three regions: many, medium, and few-shot regions. These regions are defined based on the number of training samples per label in each discretized bin, with 'Few' having fewer than 20 samples, 'Median' between 20 and 100 samples, and 'Many' exceeding 100 samples.

For each test sample $\mathbf{x}_{query}$, we retrieve its $k$ nearest neighbors $(\mathbf{x}_1, ..., \mathbf{x}_k)$ from the training set. Throughout the document, we use cosine similarity as a measure for retrieving neighboring points to the query point. Figure 2 empirically validates on the two datasets that in-context learning can perform better than simply averaging the context labels. Interestingly, for the few shot regions in both datasets, we observe that the empirical error of averaging increases rapidly with the number of context examples. This underscores the challenge of retrieving neighboring samples that are most useful for prediction in this region, even when

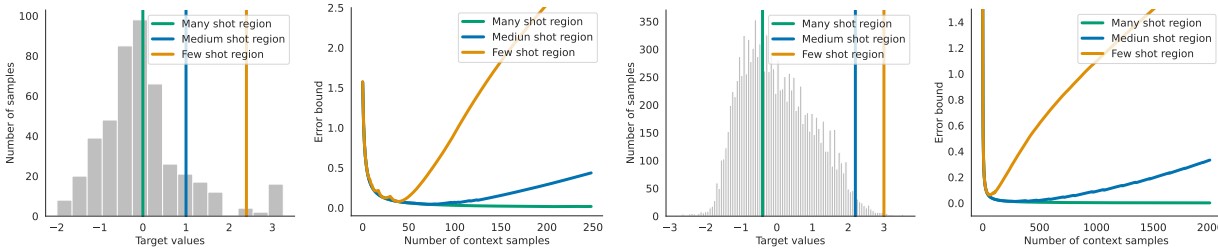

(a) Boston dataset distribution and Error bound   (b) Age dataset distribution and Error bound

Figure 1: Training distribution of two datasets: Boston (a) and AgeDB (b). The empirical expectations of errors are computed assuming ideal retrieval of neighbors for a new query sample (i.e., retrieving samples that are semantically relevant in terms of the target variable, rather than just close in the input space). The behavior of examples from different shot regions varies distinctly with the number of context examples. In the many-shot regions, the error bound stabilizes as more examples are provided, whereas in the few-shot regions, additional context examples lead to an increase in the error bound.

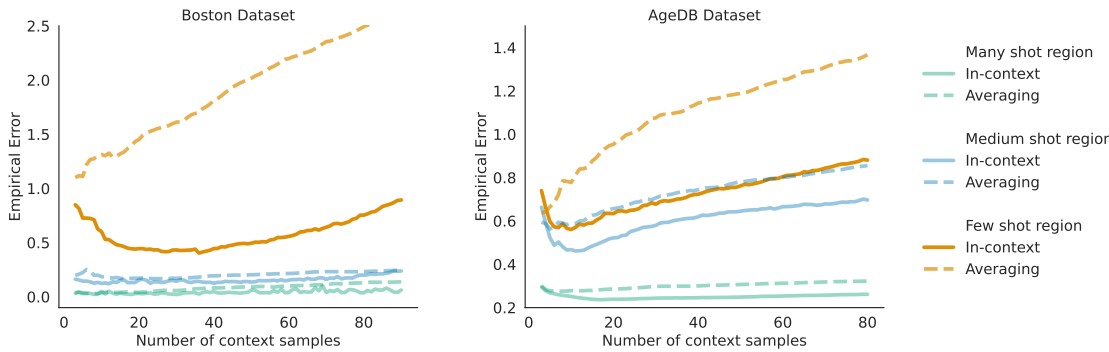

Figure 2: Empirical Error: Averaging vs In-context learning (using GPT-2 model from (Garg et al., 2022)).

using a small number of context examples. As highlighted by Steck et al. (2024), samples that are close in feature space according to a given distance metric (e.g., cosine similarity) do not necessarily have similar target values. In imbalanced datasets, this discrepancy between proximity in feature space and similarity in target values makes it difficult to select representative samples for the region, as the difference in density makes it more likely to retrieve samples from the majority region.

### 3.3 Retrieving neighbours

Retrieving representative samples as context in practice is challenging, particularly for imbalanced datasets, because the probability of retrieving samples from the majority region is higher. To address this challenge, we propose creating a second training set, where the number of samples in each region is inversely proportional to its representation in the original set — majority regions have fewer points, while minority regions are overrepresented, we refer to this dataset as inverse density as seen in Figure 5 in the appendix. For each new query sample, we retrieve $k = k'_s + \tilde{k}_s$ neighboring examples from both (1) the training set ($k'_s$ neighbors) and (2) and inverse density dataset ($\tilde{k}_s$ neighbors), to ensure balanced representation and reduce the risk of bias toward the majority region. We refer to this version as augmented. In Figure 3, we compare the performance

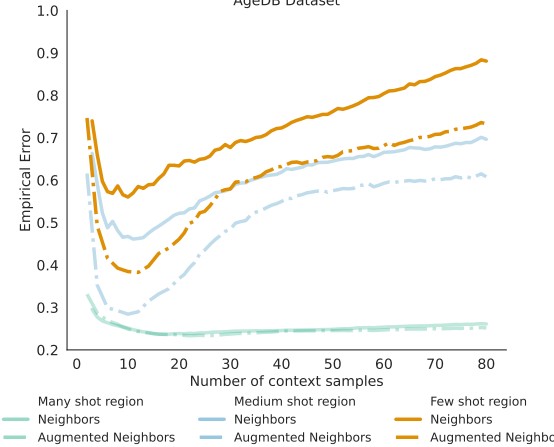

Figure 3: Impact of neighbors retrieval on performances (using GPT-2 model proposed in (Garg et al., 2022)).

of in-context learning by retrieving neighboring context examples from the original training dataset and the augmented version. We can clearly see that this strategy decreases the error on both medium and few-shot regions, and accentuates the U-shape curve, as expected theoretically. Other resampling strategies, such as Smoter and undersampling, are discussed in the ablation.

## 4 EXPERIMENTS

### 4.1 Experimental Setup

**Datasets**: We use three benchmark datasets curated by Yang et al. (2021) specifically for imbalanced regression tasks: AgeDB-DIR, derived from the AgeDB dataset for age estimation Moschoglou et al. (2017); IMDB-WIKI-DIR, another age estimation dataset sourced from IMDB-WIKI Rothe et al. (2018); and STS-B-DIR, which measures text similarity between two sentences based on the Semantic Textual Similarity Benchmark Wang et al. (2018). Additionally, we use six tabular datasets, namely Boston Harrison & Rubinfeld (1978), Concrete Yeh (2007), Abalone Nash et al. (1995), Communities Redmond (2009), Kin8nm, and an engineering design dataset: Airfoil Chen et al. (2019), with inherent imbalances. For these tabular datasets, we created a balanced test set to evaluate model performance, to assess model performance in scenarios with a small number of training samples, different feature sizes, and varying levels of imbalance.

**Evaluation Metrics**: Consistent with established protocols by Yang et al. (2021), we assess model performance using Mean Absolute Error (MAE) and Geometric Mean (GM) for the AgeDB-DIR and IMDB-WIKI-DIR datasets. For the other datasets, we use Mean Squared Error (MSE) as the primary metric, consistent with Yang et al. (2021). We present our results across four predefined shot regions—All, Many, Median, and Few—which categorize the subsets of the datasets based on the number of training samples available per label within each discretized bin. Specifically, the 'Few' category includes bins with fewer than 20 samples, 'Median' encompasses bins with 20 to 100 samples, and 'Many' refers to bins with over 100 samples per label.

**Preprocessing of inputs for In-Context Learning**: In this paper, we propose an in-context learning approach as opposed to in-weight learning, which means we do not train any models directly. For the benchmark datasets, we preprocess images and text into embeddings using the Hugging Face implementation of CLIP (Radford et al., 2021) and BERT ("all-mpnet-base-v2" model) (Reimers & Gurevych, 2019). Specifically, for each image from the training and testing datasets, we extract embeddings from the CLIP image encoder before pooling, resulting in 768-dimensional features. A similar process is applied to the STS-B-DIR dataset, where we use BERT's text embeddings for each sentence, also resulting in 768-dimensional features. For the tabular datasets, we use the feature representations directly as inputs to the transformer. Moreover, we consider two in-context learning models: the first from Garg et al. (2022), referred to as GPT2, which uses the GPT-2 architecture and is trained on non-linear data points; the second model from Müller et al. (2023), referred to as Prior Fitted Networks (PFN). They accept 20-dimensional and 18-dimensional inputs, respectively. When the number of dimensions exceeds the model's input size, we split the input feature representation into non-overlapping chunks and ensemble the predictions from each chunk. To ensure the robustness and reproducibility of our results, we conducted three separate experiments using different random seeds, between 0 and 3. We report the average results over all three seeds to account for the randomness introduced by inverse density sampling for the second training set. In all experiments, for the GPT2 model from Garg et al. (2022), we retrieve $k'_s = \tilde{k}_s = 10$ nearest neighbors, for the PFN model we retrieve $k'_s = \tilde{k}_s = 15$, except for the IMDB dataset where $k'_s$ was equal to 5. This choice is motivated by the preliminary experiments conducted in section 3.3 as observed in Figure 3 where the minimum error in all regions is achieved for 10 neighbors. A sensitivity analysis on the number of neighbors is presented in Table 9. An NVIDIA RTX 2080 GPU was used for all the experiments.

### 4.2 Experimental Results

**AgeDB:** In Table 1, we report the results obtained on the AgeDB-DIR benchmarks. We compared the best combination of previously reported results to our approach. Given the small size of the training dataset (12.2K images), we observe that in-context learning outperforms all in-weight learning baselines in all shot regions,

surprisingly even in the many-shot region. In the few-shot setting, our localized version using the Garg et al. (2022) model achieves the lowest mean absolute error of 7.83, which is an improvement of 1.4 error points over the combination ConR+LDS+FDS+Ranksim proposed in Keramati et al. (2024). Furthermore, on the AgeDB dataset, in-weight learning achieves the highest performance on all eight shot-metrics combinations with the best MAE results of 6.05 on the overall test set.

Table 1: Main results for AgeDB-DIR. Best results are in **bold** and second-best results are underlined.

| | Learning | | MAE ↓ | | | | GM ↓ | | | |
| | IWL | ICL | | | | | | | | |
| **Metrics** | | | | | | | | | | |
| Shot | | | all | many | medium | few | all | many | medium | few |
| MSE loss | ✓ | ✗ | 7.77 | 6.62 | 9.55 | 13.67 | 5.05 | 4.23 | 7.01 | 10.75 |
| LDS + FDS (Yang et al., 2021) | ✓ | ✗ | 7.55 | 7.01 | 8.24 | 10.79 | 4.72 | 4.36 | 5.45 | 6.79 |
| RankSim (Gong et al., 2022) | ✓ | ✗ | 7.02 | 6.49 | 7.84 | 9.68 | 4.53 | 4.13 | 5.37 | 6.89 |
| VIR (Wang & Wang, 2024) | ✓ | ✗ | 6.99 | 6.39 | 7.47 | 9.51 | 4.41 | 4.07 | 5.05 | 6.23 |
| ConR + LDS + FDS + RankSim (Keramati et al., 2024) | ✓ | ✗ | 6.81 | 6.32 | 7.45 | 9.21 | 4.39 | 3.81 | 5.01 | 6.02 |
| PFN - localized (Müller et al., 2023) (Ours) | ✗ | ✓ | 6.58 | **5.61** | 8.49 | 10.49 | 4.29 | **3.58** | 6.30 | 8.19 |
| GPT2 - localized (Garg et al., 2022) (Ours) | ✗ | ✓ | **6.05** | 5.67 | **6.71** | **7.83** | **3.79** | 3.59 | **4.17** | **4.90** |

Table 2: Main results for IMDB-WIKI-DIR. Best results are in **bold** and second-best results are underlined.

| | Learning | | MAE ↓ | | | | GM ↓ | | | |
| | IWL | ICL | | | | | | | | |
| **Metrics** | | | | | | | | | | |
| Shot | | | all | many | medium | few | all | many | medium | few |
| MSE loss | ✓ | ✗ | 8.06 | 7.23 | 15.12 | 26.33 | 4.57 | 4.17 | 10.59 | 20.46 |
| LDS + FDS (Yang et al., 2021) | ✓ | ✗ | 7.78 | 7.20 | 12.61 | 22.19 | 4.37 | 4.12 | 7.39 | 12.61 |
| RankSim + FDS (Gong et al., 2022) | ✓ | ✗ | 7.35 | 6.81 | 11.50 | 22.75 | 4.05 | 3.85 | 6.05 | 14.68 |
| ConR + FDS (Keramati et al., 2024) | ✓ | ✗ | 7.29 | 6.90 | 12.01 | 21.72 | 4.02 | 3.83 | 6.71 | 12.59 |
| VIR (Wang & Wang, 2024) | ✓ | ✗ | **7.19** | **6.56** | 11.81 | 20.96 | **3.85** | **3.63** | 6.51 | 12.23 |
| PFN - localized (Müller et al., 2023) (Ours) | ✗ | ✓ | 8.96 | 8.71 | **10.79** | **16.33** | 5.26 | 5.17 | **6.00** | **9.42** |
| GPT2 - localized (Garg et al., 2022) (Ours) | ✗ | ✓ | 7.76 | 7.35 | 11.15 | 17.71 | 4.29 | 4.13 | 5.96 | 11.00 |

**IMDB:** In Table 2, we report the results obtained on the IMDB-WIKI-DIR benchmarks. This is the largest dataset used in this study with 191.5K training samples. Both models (Garg et al., 2022; Müller et al., 2023) using our localized approach achieve the lowest error in the few-shot and medium-shot regions. The best-performing in-context model is (Müller et al., 2023), which achieves an MAE of 16.33 and 10.79, improving over state-of-the-art methods on age estimation by 34.63 and 1.02 error points, respectively. In the many-shot region, in-weight learning outperforms in-context learning due to the abundance of training samples (almost 150k samples), which facilitates more effective learning and better generalization.

**STS:** Table 3 presents the results on the STS-B-DIR dataset. In this text modality, both models using our localized method consistently and substantially improve the results over state-of-the-art methods in all regions, achieving the best MSE of 0.528 on the overall test set and 0.566 in the few-shot region.

**Tabular:** Table 4 shows the average results on six tabular datasets. While the previous results on the previous datasets rely on features extracted from pre-trained models, a direct comparison of in-weight learning vs in-context learning can be made using tabular datasets. The results clearly show that in-context learning outperforms

Table 3: Main results for STS-B-DIR. Best results are in **bold**.

| **Metrics** | MSE ↓ | | | |
| Shot | all | many | medium | few |
| MSE loss | 0.974 | 0.851 | 1.520 | 0.984 |
| LDS + FDS | 0.903 | 0.806 | 1.323 | 0.936 |
| VIR | 0.892 | 0.795 | 0.899 | 0.780 |
| RankSim | 0.865 | 0.876 | 0.867 | 0.670 |
| PFN - localized | 0.544 | 0.536 | 0.547 | 0.618 |
| GPT2 - localized | **0.528** | **0.524** | **0.527** | **0.566** |

different machine learning methods in the medium and few shot regions. Across the six datasets the proposed method using (Garg et al., 2022) model is almost consistently ranked first across all those datasets on the few shot region, and is highly competitive in comparison to other methods in the medium shot region.

Table 4: Average results on the Tabular datasets. Best results are in **bold**. (med.: medium)

| | Learning IWL ICL | | RMSE ↓ | | | | Rank ↓ | | | |
|---|---|---|---|---|---|---|---|---|---|---|
| Metrics | | | | | | | | | | |
| Shot | | | all | many | medium | few | all | many | med. | few |
| KNN | ✗ ✓ | | 3.96 ± 0.23 | 1.72 ± 0.30 | 3.07 ± 0.48 | 6.41 ± 0.47 | 6.2 | 4.5 | 6.0 | 6.5 |
| Linear Regression | ✗ ✓ | | 4.00 ± 0.25 | 2.10 ± 0.24 | 3.28 ± 0.47 | 6.18 ± 0.54 | 5.3 | 5.3 | 6.2 | 5.5 |
| Decision Tree | ✓ ✗ | | 3.03 ± 0.32 | 1.90 ± 0.42 | 2.72 ± 0.63 | 4.29 ± 0.48 | 7.0 | 7.3 | 7.2 | 5.5 |
| Gradient Boosting | ✓ ✗ | | 2.50 ± 0.07 | 1.26 ± 0.21 | 2.04 ± 0.17 | 4.05 ± 0.13 | 3.8 | 3.2 | 3.8 | 5.0 |
| Random Forest | ✗ ✗ | | 2.57 ± 0.16 | **1.22 ± 0.35** | 2.07 ± 0.30 | 4.16 ± 0.35 | 3.5 | **2.5** | 3.5 | 4.3 |
| Neural Networks | ✓ ✗ | | 2.49 ± 0.21 | 1.44 ± 0.25 | 2.13 ± 0.21 | 3.79 ± 0.51 | 2.7 | 3.2 | 3.2 | 2.6 |
| PFN - localized (Ours) | ✗ ✓ | | 2.67 ± 0.19 | 1.37 ± 0.23 | 2.03 ± 0.27 | 4.38 ± 0.26 | 5.2 | 4.5 | 3.3 | 5.3 |
| GPT2 - localized (Ours) | ✗ ✓ | | **2.34 ± 0.19** | 1.72 ± 0.35 | **1.95 ± 0.22** | **3.31 ± 0.39** | **2.3** | 5.5 | **2.8** | **1.2** |

## 5 ABLATIONS

**Representation learning.** In our experiment with the age datasets, we used CLIP embeddings across all age datasets to assess the effectiveness of in-context learning compared to in-weight learning. The methods in Table 1 were trained to learn representations directly from the images, whereas we used pre-extracted CLIP embeddings. We trained a three-layer MLP with 256 neurons in the hidden layers using these embeddings. Additionally, we conducted an experiment where we fine-tuned the GPT2 model on the AgeDB dataset. Results presented in Table 5 indicate that also in this scenario in-context learning consistently outperforms in-weight learning. Interestingly, the fine-tuned GPT2 model achieves almost similar performance to the in-context learning approach. These results suggest that in-context learning is a particularly effective solution for relatively small datasets with the presence of scarce data, such as AgeDB.

Table 5: Ablation results for AgeDB-DIR benchmark.

| | Learning IWL ICL | | MAE ↓ | | | | GM ↓ | | | |
|---|---|---|---|---|---|---|---|---|---|---|
| Metrics | | | | | | | | | | |
| Shot | | | all | many | medium | few | all | many | medium | few |
| MLP | ✓ ✗ | | 6.67 | 5.97 | 8.22 | 9.01 | 4.29 | 3.86 | 5.48 | 5.84 |
| GPT2 - Finetuned (Garg et al., 2022) | ✓ ✗ | | 6.14 | 5.73 | **6.69** | 8.09 | 4.01 | 3.70 | 4.59 | 5.39 |
| GPT2 - localized (Garg et al., 2022) (Ours) | ✗ ✓ | | **6.05** | **5.67** | 6.71 | **7.83** | **3.79** | **3.59** | **4.17** | **4.90** |

**All Context vs. Localized Approach.** A key rationale for using the PFN model from Müller et al. (2023) is its lack of positional embedding. This allows for a direct comparison of the model's performance using the entire training set versus a localized version as context. Figure 4 presents results across the first five tabular datasets. The engineering dataset, which has more than 30k training samples, could not fit in memory and was therefore excluded from this ablation. The results show a significant performance improvement with the localized approach in in-context learning compared to the non-localized version across all regions, with a particularly significant improvement in the few-shot region.

**Sampling strategy:** In Section 3.3, we presented a strategy to mitigate the underrepresentation of minority points when retrieving neighbors. In this section, we compare different sampling strategies on the tabular dataset. The 'Vanilla' approach retrieves neighbors from the original training set. 'Downsampling' reduces the majority region, creating a balanced training set and retrieving neighbors only from this balanced set. SMOTER (Branco et al., 2017) generates new synthetic data points for the minority region, and neighbors

are retrieved exclusively from this augmented dataset. 'Inverse' uses a dataset where the number of samples in each region is inversely proportional to its representation in the original set. Lastly, our strategy involves creating an Augmented dataset to ensure balanced representation while preserving diversity in the majority region. As seen in Table 6, across the datasets, our method demonstrates the best trade-off for mitigating bias in all regions.

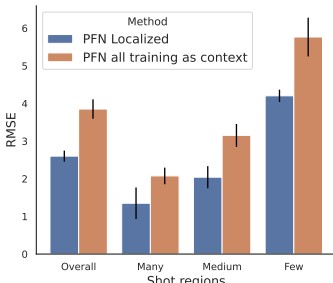

Figure 4: Localization vs all training set set as context

| Metrics | | RMSE ↓ | | |
|---|---|---|---|---|
| Shot | all | many | medium | few |
| Vanilla | 2.43 | **1.41** | 1.99 | 3.70 |
| Downsampling | 2.65 | 2.42 | 2.38 | 3.17 |
| Inverse | 2.79 | 2.93 | 2.24 | **3.13** |
| SMOTER | 2.43 | 1.68 | 2.09 | 3.37 |
| Augmented (Ours) | **2.34** | 1.72 | **1.95** | 3.31 |

Table 6: Ablation results on the Tabular datasets using alternative sampling strategy.

# 6 Conclusion

In this work, we proposed the use of in-context learning to address the challenge of imbalanced regression, where a model can learn from context examples during inference without any additional training on the model weights. We show that a single in-context learning model can adapt to multiple tasks and improve performance in regions with high imbalance, unlike in-weight regression models, which often fail to generalize in minority regions and require specific training for each task. Through theoretical and empirical analyses, we highlighted the importance of localized context—using a small number of selected neighbors—in mitigating the bias inherent in imbalanced label distributions. Our evaluation across several benchmark datasets with diverse modalities revealed that in-context learning achieves superior results in regions of high imbalance.

Our comparative analysis demonstrated that two in-context learning models (Müller et al., 2023; Garg et al., 2022) based on Transformer architectures can achieve strong performance in imbalanced regression tasks for one-dimensional labels. While models are currently trained to predict a single value for a given query, future work will focus on extending our approach to multi-dimensional labels, such as depth estimation. Future work could also explore other model variants, and further investigate the factors contributing to performance differences between different in-context learning models for minority and majority regions.

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

## A  Appendix

## B  Retrieving neighbours

In this section, we further motivate the design choice of our augmented strategy. As shown in Figure 5, in the case of imbalanced regression, retrieving neighbors from the original training set can result in selecting points from the same region, which may not provide informative insights into local feature behavior. From Figure 5: Using the vanilla distribution will skew the choice of neighbours towards the majority region. Conversely, with 'inverse sampling', the distribution becomes more representative of the minority region. The aim of our augmented strategy is to find a good trade-off between these two sampling methods. Conversely, our strategy makes it easier to retrieve diverse and informative neighbors around the query point $x_{query}$.

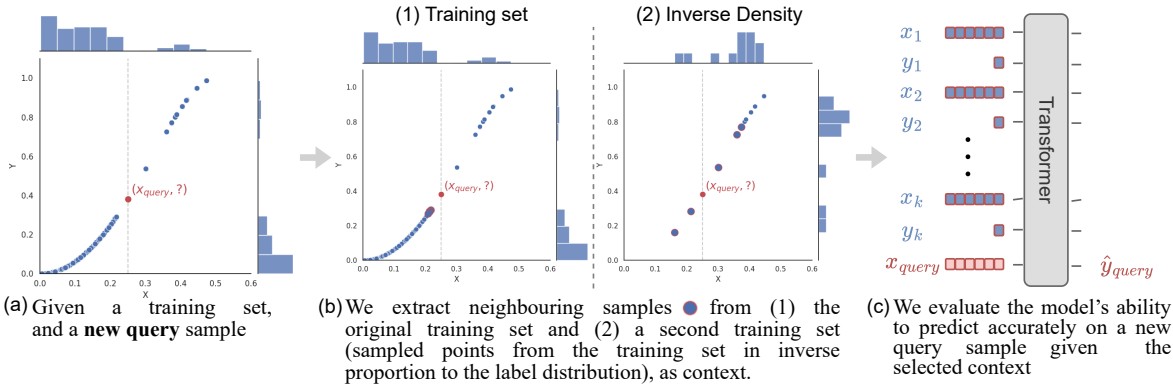

(a) Given a training set, and a **new query** sample

(b) We extract neighbouring samples ● from (1) the original training set and (2) a second training set (sampled points from the training set in inverse proportion to the label distribution), as context.

(c) We evaluate the model's ability to predict accurately on a new query sample given the selected context

Figure 5: An overview of the proposed approach for imbalanced regression. Rather than relying on in-weight learning, which trains models directly on the training data, we propose leveraging in-context learning. For each query sample, we retrieve $k = k'_s + \tilde{k}_s$ neighboring examples from both (1) the training set ($k'_s$ neighbors) and (2) and inverse density dataset ($\tilde{k}_s$ neighbors), where the number of samples in each region is inversely proportional to its representation in the original set, and feed these as context to the model. This serves a dual purpose: avoiding bias toward the mean of the training set, which is crucial for tail regions, and reducing the memory requirement of the transformer.

## C  Experiment Details

### C.1  Datasets

**Age Estimation**  We evaluated our method using two imbalanced regression benchmarks for age estimation provided by Yang et al. (2021): IMDB-WIKI-DIR and AgeDB-DIR. The IMDB-WIKI dataset (Rothe et al., 2018) includes 191,509 images for training and 11,022 images for validation and testing. The ages are discretized into 1-year bins, from age 0 to 186. The AgeDB dataset (Moschoglou et al., 2017) contains 16,488 samples. AgeDB-DIR was structured similarly to IMDB-WIKI-DIR, with age bins ranging from 0 to 101.

**Text Similarity**  We evaluated our method using imbalanced regression benchmarks for textual similarity estimation provided by Yang et al. (2021). STS-B-DIR (Wang et al., 2018) consists of sentence pairs sourced from news headlines, video and image captions, and natural language inference data. Each pair is annotated by different annotators, resulting in an averaged continuous similarity score ranging from 0 to 5. The task is to predict these similarity scores based on the sentence pairs. The training dataset includes 5,249 pairs of sentences, with validation and test sets containing 1,000 pairs each.

**Tabular Datasets**  We used five datasets from the UCI machine learning repository (Frank, 2010): Boston, Concrete, Abalone, Kin8nm, and Communities, as a sixth dataset from engineering: the airfoil dataset. The label distributions for all the tabular datasets is shown in Figure 7.

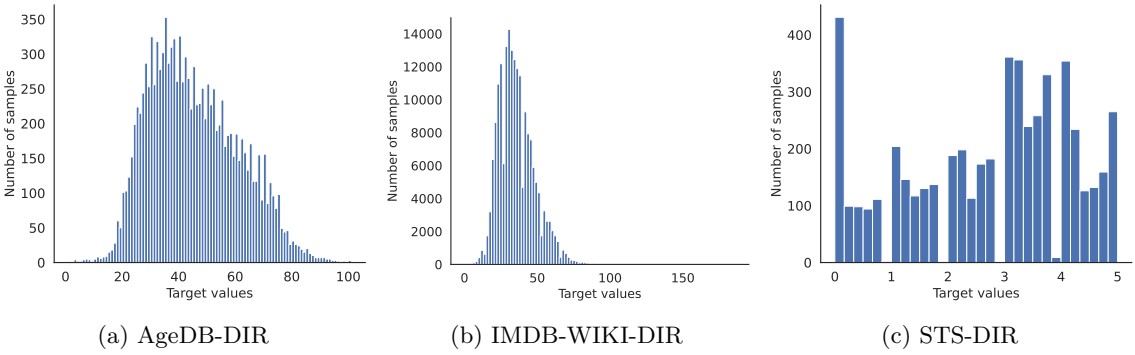

Figure 6: Training distribution of labels for the image and text modality.

For the UCI datasets, we created training and testing sets, ensuring the test sets were balanced, using the Algorithm **??**. Due to its small size, the Boston dataset was split 90% for training and 10% for testing, and the bin size was set to 15. The other datasets were split 80% for training and 20% for testing and the bin size was set to 50. The specific details:

- Boston: 13 features, 462 training samples, 44 testing samples.

- Concrete: 8 features, 836 training samples, 194 testing samples.

- Abalone: 7 features, 8,634 training samples, 543 testing samples.

- Kin8nm: 8 features, 6,766 training samples, 1,426 testing samples.

- Communities: 101 features, 1,684 training samples, 310 testing samples.

For the engineering dataset: airfoil dataset (Chen et al., 2019), we used 96 Bézier features (Heyrani Nobari et al., 2021), with 38,802 training samples and 9,701 testing samples.

### C.2   Metrics

In the main paper, we reported results using the Mean Absolute Error (MAE), Root Mean Squared Error (RMSE), and Geometric Mean (GM). The formulas for these metrics are as follows:

$$\text{MAE} = \frac{1}{N}\sum_{i=1}^{N}|y_i - \hat{y}_i|, \quad \text{RMSE} = \sqrt{\frac{1}{N}\sum_{i=1}^{N}(y_i - \hat{y}_i)^2}, \quad \text{GM} = \left(\prod_{i=1}^{N}|y_i - \hat{y}_i|\right)^{\frac{1}{N}}$$

For the $i$-th sample, $y_i$ is the actual value, $\hat{y}_i$ is the predicted value, and $N$ is the number of samples. Lower values of MAE, RMSE, and GM indicate better predictive accuracy.

### C.3   In-context learning model Details

The model from Garg et al. (2022) uses architectures from the GPT-2 decoder transformer with a learnable positional embedding The PFN model from Müller et al. (2023) employs a standard encoder transformer, without positional embedding. The detailed architecture is as follows:

### C.4   Implementation Details

For the tabular datasets, we used baselines from scikit-learn (Pedregosa et al., 2011). Specifically, for K-Nearest Neighbors (KNN), we reported results using 10 neighbors, akin to our localized method. For

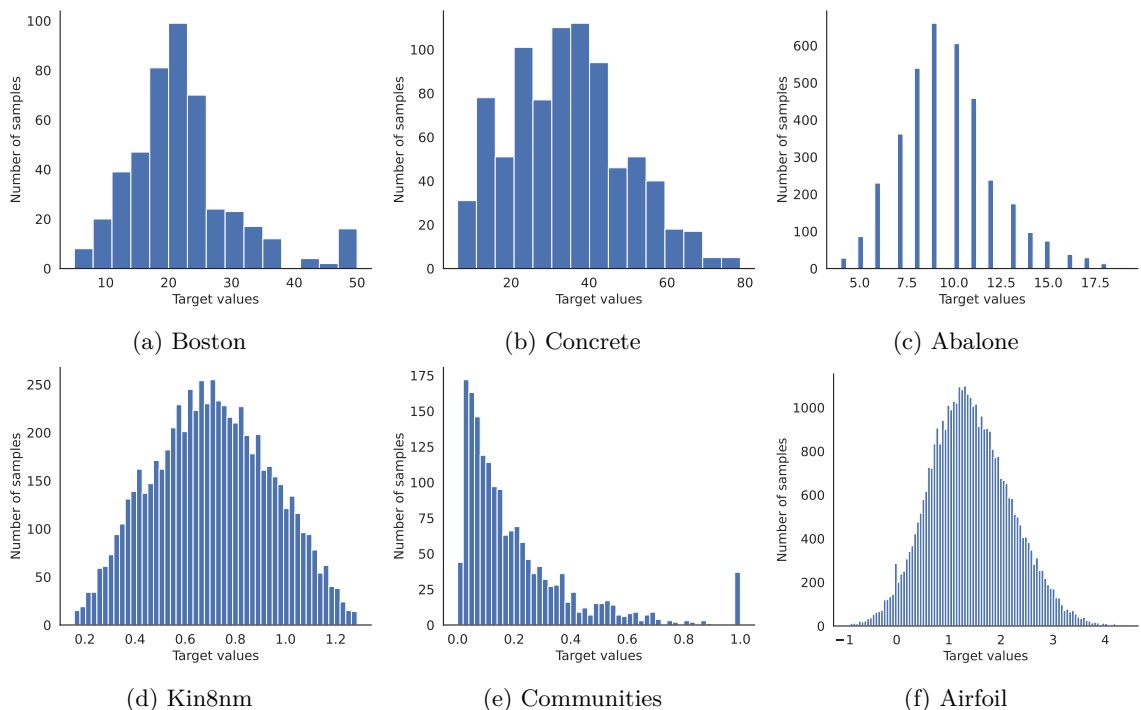

Figure 7: Training distribution of labels for the tabular datasets.

Table 7: Architecture details of GPT2 and PFN models.

| Model | Input Dimension | Embedding Size | Number of Layers | Number of Heads |
|-------|-----------------|----------------|------------------|-----------------|
| GPT2  | 20              | 256            | 12               | 8               |
| PFN   | 18              | 512            | 6                | 8               |

Decision Trees, Gradient Boosting, Linear Regression, Random Forest and Neural Networks, we used the default parameters.

For IMDB-WIKI-DIR and AgeDB-DIR, the baseline uses a ResNet-50 backbone trained from scratch. For STSB-DIR, the baseline uses a BiLSTM with GloVe word embeddings. The details on this datasets can be found in Yang et al. (2021).

Standard scaling was applied to the input features for these models. For in-context learning, we applied both standard scaling and a power transform to the features and then concatenated the two representations.

## C.5 Computational Analysis

We conducted an analysis comparing in-weight learning and in-context learning approaches in terms of training/processing time, inference time, and memory requirements. Table 8 summarizes our findings for the IMDB-Wiki dataset. For in-context learning, the training time is effectively zero, but we've included the time to process and extract features from the training set (189,794 images) using CLIP. The inference time includes both feature extraction and prediction. These results show that while the in-context learning method has a longer inference time, it requires less memory and significantly reduces the upfront training/processing time.

Table 8: In-weight vs. In-context learning time analysis on IMDB-Wiki

| Method | Train/Process Time | Inference Time | GPU Memory |
|---|---|---|---|
| In-weight learning (ResNet-50) | 8:12:00 | 0.01s | 578M |
| In-context learning (GPT2 model) | 2:12:33 | 0.06s | 506M |

This trade-off may be particularly beneficial in scenarios where quick adaptation to new tasks is valuable, or when computational resources for extensive training are limited.

## D   Ablations

### D.1   Sensitivity to context length size

In this section, we conduct a sensitivity analysis of the model to different context lengths. As shown in Table 9, the averaged results across the six tabular datasets indicate consistent performance for 'small' number of neighbors.

Table 9: Sensitivity to the number of neighbors: Average results on tabular datasets

| Metrics | RMSE $\downarrow$ | | | |
|---|---|---|---|---|
| Shot | all | many | medium | few |
| PFN - localized $\tilde{k}_s = k'_s = 5$ | 2.66 | 1.46 | 2.08 | 4.21 |
| PFN - localized $\tilde{k}_s = k'_s = 10$ | 2.72 | 1.43 | 2.05 | 4.45 |
| PFN - localized $\tilde{k}_s = k'_s = 15$ | 2.68 | 1.38 | 2.03 | 4.38 |
| PFN - localized $\tilde{k}_s = k'_s = 20$ | 2.66 | 1.38 | 2.02 | 4.39 |
| GPT2 - localized $\tilde{k}_s = k'_s = 5$ | 2.46 | 1.82 | 2.06 | 3.47 |
| GPT2 - localized $\tilde{k}_s = k'_s = 10$ | 2.35 | 1.72 | 1.95 | 3.31 |
| GPT2 - localized $\tilde{k}_s = k'_s = 15$ | 2.32 | 1.94 | 1.89 | 3.20 |
| GPT2 - localized $\tilde{k}_s = k'_s = 20$ | 2.40 | 1.92 | 2.03 | 3.32 |

## E   Results

In this section, we report the average results and standard deviations for all nine datasets used in the paper. These results are obtained by running experiments with three different random seeds: 0, 1, and 2. For clarity, neighbor selection is performed deterministically using the sklearn KNN implementation.

### E.1   AgeDB-DIR

The complete results with the standard deviations for the AgeDB dataset are shown in Table 10 .

Table 10: Main results for AgeDB-DIR benchmark.

| Metrics | MAE $\downarrow$ | | | | GM $\downarrow$ | | | |
|---|---|---|---|---|---|---|---|---|
| Shot | all | many | medium | few | all | many | medium | few |
| PFN - localized | $6.58 \pm 0.00$ | $5.61 \pm 0.01$ | $8.49 \pm 0.00$ | $10.49 \pm 0.06$ | $4.29 \pm 0.02$ | $3.58 \pm 0.02$ | $6.30 \pm 0.07$ | $8.19 \pm 0.06$ |
| GPT2 - localized | $6.05 \pm 0.03$ | $5.67 \pm 0.05$ | $6.71 \pm 0.01$ | $7.83 \pm 0.02$ | $3.79 \pm 0.07$ | $3.59 \pm 0.11$ | $4.17 \pm 0.08$ | $4.90 \pm 0.08$ |

### E.2   IMDB-WIKI-DIR

The complete results with the standard deviations for the IMDB-WIKI dataset are shown in Table 11 .

Table 11: Main results for IMDB-WIKI-DIR.

| Metrics | MAE ↓ | | | | GM ↓ | | | |
|---|---|---|---|---|---|---|---|---|
| Shot | all | many | medium | few | all | many | medium | few |
| PFN - localized | 8.96 ± 0.02 | 8.71 ± 0.02 | 10.79 ± 0.10 | 16.33 ± 0.12 | 5.26 ± 0.02 | 5.17 ± 0.01 | 6.00 ± 0.12 | 9.42 ± 0.22 |
| GPT2 - localized | 7.76 ± 0.05 | 7.35 ± 0.04 | 11.15 ± 0.06 | 17.71 ± 0.25 | 4.29 ± 0.06 | 4.13 ± 0.06 | 5.96 ± 0.07 | 11.00 ± 0.35 |

## E.3 STS-B-DIR

The complete results with the standard deviations for the STS-B-DIR dataset are shown in Table 12 .

Table 12: Results for STS-B-DIR.

| Metrics | MSE ↓ | | | |
|---|---|---|---|---|
| Shot | all | many | medium | few |
| PFN - localized | 0.544 ± 0.006 | 0.536 ± 0.008 | 0.547 ± 0.027 | 0.618 ± 0.016 |
| GPT2 - localized | 0.528 ± 0.002 | 0.524 ± 0.005 | 0.527 ± 0.019 | 0.566 ± 0.008 |

## E.4 Tabular datasets

In this section we report individually the results for each of the tabular datasets used. The results for Boston (Harrison & Rubinfeld, 1978) are in Table 13. The results for Concrete (Yeh, 2007) are in Table 14. The same applied for Abalone (Nash et al., 1995) in Table 15 , Communities (Redmond, 2009) in Table 17, Kin8nm in Table 16, and an engineering design dataset: Airfoil (Chen et al., 2019) in Table 18.

Table 13: Average results on the Boston dataset.

| | Learning | | RMSE ↓ | | | |
|---|---|---|---|---|---|---|
| Metrics | IWL | ICL | | | | |
| Shot | | | all | many | medium | few |
| Knn | ✗ | ✓ | 6.32 ± 0.90 | 1.60 ± 0.80 | 3.77 ± 1.02 | 9.39 ± 1.56 |
| Decision Tree | ✓ | ✗ | 5.40 ± 1.53 | 2.71 ± 0.71 | 5.39 ± 2.62 | 5.83 ± 0.70 |
| Gradient Boosting | ✓ | ✗ | **3.39 ± 0.23** | **1.19 ± 0.16** | **2.77 ± 0.46** | 4.50 ± 0.01 |
| Neural Networks | ✓ | ✗ | 3.41 ± 0.89 | 2.04 ± 0.74 | 2.92 ± 0.77 | 4.23 ± 1.51 |
| PFN - localized (Ours) | ✗ | ✓ | 3.90 ± 0.71 | 1.43 ± 0.55 | 2.84 ± 0.80 | 5.41 ± 1.11 |
| GPT2 - localized (Ours) | ✗ | ✓ | 3.52 ± 0.91 | 2.35 ± 1.59 | 3.21 ± 0.60 | **4.06 ± 1.61** |

Table 14: Average results on the Concrete dataset.

| | Learning | | RMSE ↓ | | | |
|---|---|---|---|---|---|---|
| Metrics | IWL | ICL | | | | |
| Shot | | | all | many | medium | few |
| Knn | ✓ | ✗ | 12.39 ± 0.44 | 6.65 ± 0.92 | 10.36 ± 1.67 | 19.91 ± 1.14 |
| Decision Tree | ✓ | ✗ | 7.47 ± 0.23 | 5.78 ± 1.75 | 6.29 ± 0.72 | 10.93 ± 1.88 |
| Gradient Boosting | ✓ | ✗ | 6.58 ± 0.14 | **4.39 ± 1.07** | 5.13 ± 0.38 | 10.66 ± 0.71 |
| Neural Network | ✓ | ✗ | 6.99 ± 0.28 | 4.84 ± 0.68 | 6.18 ± 0.27 | 10.43 ± 1.53 |
| PFN - localized (Ours) | ✓ | ✗ | 7.02 ± 0.39 | 4.57 ± 0.73 | **4.99 ± 0.69** | 11.79 ± 0.25 |
| GPT2 - localized (Ours) | ✓ | ✗ | **5.85 ± 0.09** | 4.54 ± 0.35 | 5.16 ± 0.48 | **8.31 ± 0.41** |

Table 15: Average results on the Abalone dataset.

| | Learning IWL ICL | | RMSE ↓ | | | |
|---|---|---|---|---|---|---|
| **Metrics** | | | | | | |
| Shot | | | all | many | medium | few |
| Knn | ✓ ✗ | | $4.39 \pm 0.03$ | $1.57 \pm 0.07$ | $3.42 \pm 0.19$ | $7.57 \pm 0.08$ |
| Decision Tree | ✓ ✗ | | $4.48 \pm 0.14$ | $2.28 \pm 0.07$ | $3.66 \pm 0.38$ | $7.33 \pm 0.28$ |
| Gradient Boosting | ✓ ✗ | | $4.33 \pm 0.04$ | $1.51 \pm 0.03$ | $3.42 \pm 0.17$ | $7.47 \pm 0.05$ |
| Neural Network | ✓ ✗ | | $\mathbf{4.11 \pm 0.05}$ | $\mathbf{1.50 \pm 0.07}$ | $3.17 \pm 0.21$ | $7.10 \pm 0.03$ |
| PFN - localized (Ours) | ✓ ✗ | | $4.54 \pm 0.01$ | $1.83 \pm 0.09$ | $3.60 \pm 0.13$ | $7.72 \pm 0.12$ |
| GPT2 - localized (Ours) | ✓ ✗ | | $\underline{4.16 \pm 0.12}$ | $2.99 \pm 0.18$ | $\mathbf{2.80 \pm 0.24}$ | $\mathbf{6.59 \pm 0.32}$ |

Table 16: Average results on the Kin8nm dataset.

| | Learning IWL ICL | | RMSE ↓ | | | |
|---|---|---|---|---|---|---|
| **Metrics** | | | | | | |
| Shot | | | all | many | medium | few |
| Knn | ✓ ✗ | | $0.17 \pm 0.01$ | $0.11 \pm 0.01$ | $0.19 \pm 0.01$ | $0.28 \pm 0.01$ |
| Decision Tree | ✓ ✗ | | $0.24 \pm 0.01$ | $0.20 \pm 0.01$ | $0.25 \pm 0.02$ | $0.32 \pm 0.01$ |
| Gradient Boosting | ✓ ✗ | | $0.24 \pm 0.01$ | $0.16 \pm 0.01$ | $0.27 \pm 0.01$ | $0.37 \pm 0.01$ |
| Neural Network | ✓ ✗ | | $\mathbf{0.09 \pm 0.01}$ | $\mathbf{0.07 \pm 0.01}$ | $\mathbf{0.09 \pm 0.01}$ | $\mathbf{0.14 \pm 0.01}$ |
| PFN - localized (Ours) | ✓ ✗ | | $0.18 \pm 0.01$ | $0.13 \pm 0.01$ | $0.18 \pm 0.01$ | $0.28 \pm 0.01$ |
| GPT2 - localized (Ours) | ✓ ✗ | | $\underline{0.13 \pm 0.01}$ | $\underline{0.10 \pm 0.01}$ | $\underline{0.13 \pm 0.01}$ | $\underline{0.20 \pm 0.01}$ |

Table 17: Average results on the Communities dataset.

| | Learning IWL ICL | | RMSE ↓ | | | |
|---|---|---|---|---|---|---|
| **Metrics** | | | | | | |
| Shot | | | all | many | medium | few |
| Knn | ✓ ✗ | | $0.21 \pm 0.00$ | $0.07 \pm 0.01$ | $0.12 \pm 0.00$ | $0.27 \pm 0.01$ |
| Decision Tree | ✓ ✗ | | $0.27 \pm 0.01$ | $0.10 \pm 0.02$ | $0.16 \pm 0.02$ | $0.34 \pm 0.01$ |
| Gradient Boosting | ✓ ✗ | | $0.21 \pm 0.01$ | $\mathbf{0.05 \pm 0.00}$ | $0.10 \pm 0.01$ | $0.27 \pm 0.01$ |
| Neural Network | ✓ ✗ | | $0.22 \pm 0.00$ | $0.09 \pm 0.02$ | $0.13 \pm 0.00$ | $0.28 \pm 0.00$ |
| PFN - localized (Ours) | ✓ ✗ | | $0.21 \pm 0.00$ | $0.09 \pm 0.01$ | $\mathbf{0.09 \pm 0.00}$ | $0.28 \pm 0.01$ |
| GPT2 - localized (Ours) | ✓ ✗ | | $\mathbf{0.19 \pm 0.00}$ | $0.10 \pm 0.00$ | $\underline{0.10 \pm 0.01}$ | $\mathbf{0.23 \pm 0.01}$ |

Table 18: Average results on the Airfoil dataset.

| | Learning IWL ICL | | RMSE ↓ | | | |
|---|---|---|---|---|---|---|
| **Metrics** | | | | | | |
| Shot | | | all | many | medium | few |
| Knn | ✓ ✗ | | $0.27 \pm 0.00$ | $0.26 \pm 0.00$ | $0.56 \pm 0.00$ | $1.06 \pm 0.00$ |
| Gradient Boosting | ✓ ✗ | | $0.25 \pm 0.00$ | $0.25 \pm 0.00$ | $0.54 \pm 0.00$ | $1.03 \pm 0.00$ |
| Decision Tree | ✓ ✗ | | $0.34 \pm 0.00$ | $0.33 \pm 0.00$ | $0.56 \pm 0.00$ | $0.97 \pm 0.00$ |
| Neural Network | ✓ ✗ | | $\mathbf{0.12 \pm 0.00}$ | $\mathbf{0.11 \pm 0.00}$ | $\mathbf{0.27 \pm 0.00}$ | $0.56 \pm 0.00$ |
| PFN - localized (Ours) | ✓ ✗ | | $0.21 \pm 0.00$ | $0.20 \pm 0.00$ | $0.48 \pm 0.01$ | $0.84 \pm 0.02$ |
| GPT2 - localized (Ours) | ✓ ✗ | | $0.23 \pm 0.00$ | $0.23 \pm 0.00$ | $0.28 \pm 0.01$ | $\mathbf{0.47 \pm 0.00}$ |

