# OpenReview forum: "IM-Context: In-Context Learning for Imbalanced Regression Tasks"
_TMLR — Accepted by TMLR_

### Review · Reviewer_nEPj · 2024-07-22

**Summary Of Contributions:**

This paper introduces a novel approach for imbalanced regression using in-context learning. Key contributions include:

- The paper investigates in-context learning where models use prompt sequences for predictions to tackle the problem of regression with imbalanced dataset, eliminating parameter updates.

- The paper highlights the importance of localized context in reducing bias in imbalanced regions through theoretical and empirical studies.

- The paper demonstrates superior performance of in-context learning on real-world datasets with high imbalance.

- The authors of the paper show that in-context learning mitigates overfitting in underrepresented regions.

These findings provide a robust alternative to conventional imbalanced regression methods.

**Audience:**

Yes

**Broader Impact Concerns:**

No concerns.

**Claims And Evidence:**

Yes

**Requested Changes:**

- I think the "retrieving neighbors" need more details. The subsection seems a little abrupt and disjoint from the rest of the section, in my personal opinion.
- Do we do finetuning for IWL? What if we finetune the model instead of ICL? How do results compare? If not, this might be an interesting comparison.

**Strengths And Weaknesses:**

Strengths:
- The paper tackles an important problem of regression with data imbalance. This is of interest to a lot of problems in real world scenarios.
- The empirical investigation on using in-context learning to solve the task is interesting.
- The authors of the paper seems to demonstrate the effectiveness of the proposed method.
- The paper is well written and easy to follow for the most part.
- The proposed method is technically sound.

Weaknesses:
- The proposed method is pretty simple and straight-forward.
- Based on my understanding, the theoretical derivations doesn't compare IWL vs. ICL? Any theoretical justification on why ICL is better than IWL?
- The proposed method is not consistently good. For example, the proposed method is not the best performing one for Table 2.

---

> ### Author Response · Authors · 2024-09-06
>
> # Response to Reviewer
>
> Dear Reviewer,
>
> Thank you for your insightful comments. We appreciate your feedback and have addressed your points as follows:
>
> You're correct that our theoretical derivations don't directly compare In-Weight Learning (IWL) to In-Context Learning (ICL). Unlike IWL, which learns a model for the entire label distribution, ICL adapts locally based on relevant context examples. This local focus is particularly beneficial in scenarios with scarce data. The key advantage of ICL is its ability to exploit local smoothness assumptions.
>
> 1. Regarding the motivation for our approach, we have added an important clarification about neighbor selection. As highlighted by Steck et al. [1], "neighbors selected using metrics like cosine similarity in feature space do not necessarily correspond to similar outputs." **This insight underscores the challenge in selecting truly relevant neighbors for in-context learning, especially in imbalanced datasets.**
>
> 2. We thank you for suggesting a comparison between in-weight learning and in-context learning. To address this, we conducted an additional experiment where we fine-tuned the GPT2 model on the AgeDB dataset. The results, presented in Table 6, show that in-context learning consistently outperforms in-weight learning, even in this scenario. Interestingly, the fine-tuned GPT2 model achieves almost similar performance to our in-context learning approach. **Specifically, our localized GPT2 model (in-context learning) achieves the best results in most metrics, with MAE of 6.05 overall and 7.83 in the few-shot region, compared to 6.14 and 8.09 for the fine-tuned model. These results suggest that in-context learning is particularly effective for relatively small datasets with scarce data, such as AgeDB.**

---

### Review · Reviewer_wjE2 · 2024-07-29

**Summary Of Contributions:**

Summary:
This paper introduces a novel approach to address the challenge of imbalanced label distributions in regression problems. Traditional deep learning models often struggle with generalization in regions of high imbalance due to overfitting on underrepresented data. The proposed IM-Context method leverages in-context learning, where a model conditions itself on a sequence of input-label pairs to make predictions without parameter updates. The study demonstrates that localized context is crucial for reducing bias in imbalanced areas and shows that in-context learning outperforms conventional methods, especially in minority regions, across various real-world datasets. Theoretical analysis and empirical results highlight the effectiveness of this paradigm in improving predictive performance.

**Audience:**

Yes

**Claims And Evidence:**

Yes

**Requested Changes:**

Please finish the discussion required in the **weakness**

**Strengths And Weaknesses:**

**Strength**
>- This paper explores a new learning paradigm to address the imbalance regression problems, which brings new insight into the community and highly promotes the development of long-tailed learning.
>- The authors establish a theoretical error bound for the expected error when employing in-context learning for imbalanced regression, >showing that a substantial number of context examples could theoretically approximate the actual predictive distribution. However, they also highlight a critical issue where utilizing the entire training set as context can result in model bias towards the majority regions, impairing the performance on infrequent labels.
>- Extensive experiments on various datasets demonstrate the efficiency of the proposed method.

**Weakness**

> - Although IM-Context has improved compared to competitors, it relies on pre-trained in-context learning models. This trade-off warrants further discussions, such as inference time and required memory space.
> - Please elaborate on the backbone and implemental details of competitors. Is the backbone used by competitors consistent with the IM-context method?
> - It is important to acknowledge the significant contributions of several influential studies in the realm of in-context learning [1][2], as well as those focused on long-tail learning [3][4][5], which have shaped our understanding and exploration in these fields. The authors are suggested to discuss these works in this paper.

**References**

>- [1] Birth of a Transformer: A Memory Viewpoint. NeurIPS 2023.
>- [2] The Mechanistic basis of data dependence and abrupt learning in an in-context classification task. ICLR 2024.
>- [3] Self-Supervised Aggregation of Diverse Experts for Test-Agnostic Long-Tailed Recognition. NeurIPS 2022.
>- [4] A Unified Generalization Analysis of Re-Weighting and Logit-Adjustment for Imbalanced Learning. NeurIPS 2023.
>- [5] Harnessing Hierarchical Label Distribution Variations in Test Agnostic Long-tail Recognition. ICML 2024

---

> ### Author Response · Authors · 2024-09-06
>
> # Response to Reviewer
>
> Dear Reviewer,
>
> Thank you for your thoughtful and insightful comments. We greatly appreciate the time and effort you've put into reviewing our work. We have addressed your points as follows:
>
> 1. We appreciate your point about the trade-offs involved with using pre-trained in-context learning models. To address this, we've conducted additional analyses comparing our approach with competitors in terms of training/processing time, inference time, and memory requirements. Here's a summary of our findings for the IMDB-Wiki dataset:
>
>    **For in-context learning, the training time is effectively zero, but we've included the time to process and extract features from the training set (189,794 images) using CLIP. The inference time includes both feature extraction and prediction.**
>
>    These results show that while our method has a longer inference time, it requires less memory and significantly reduces the upfront training/processing time. This trade-off may be particularly beneficial in scenarios where quick adaptation to new tasks is valuable, or when computational resources for extensive training are limited.
>
>    **These results show that while our method has a longer inference time, it requires less memory and significantly reduces the upfront training/processing time.**
>
> 2. We apologize for not providing clearer details about the backbone implementations. To clarify:
>    - For IMDB-WIKI-DIR and AgeDB-DIR, the Baseline uses a ResNet-50 backbone trained from scratch.
>    - For STSB-DIR, a BiLSTM with GloVe word embeddings is used.
>
>    **Our in-context learning method uses pre-trained models without task-specific training. Specifically, we use the pre-trained CLIP ViT model to extract features for these tasks. This approach allows us to compare the performance of in-weight learning (training a model on the specific data) with in-context learning (using only pre-trained models without task-specific training) those were already included in the main paper.**
>
> 3. We sincerely thank you for pointing out these important works. Our related work section was primarily focused on imbalanced regression. As the works you highlighted are more relevant to classification tasks, we have added them to the introduction.

---

### Review · Reviewer_cCTG · 2024-08-22

**Summary Of Contributions:**

This paper argues that in-context learning is naturally well-suited to imbalanced datasets, particularly for regression. It points out that existing work on addressing imbalanced data is on classification tasks, not regression. Starting with existing pre-trained transformer-based regression models, the paper develops an approach that uses only the nearest training samples in the context prompt (both from the original dataset and an inverse density version of it), to avoiding biasing towards majority regions. A theoretical motivation for the approach is provided, and experimental comparisons demonstrate an advantage of the proposed method over existing approaches to regression on imbalanced datasets.

**Audience:**

Yes

**Broader Impact Concerns:**

Using age estimation as a motivating domain raises potential ethical concerns. Any machine learning system can be used for unethical purposes, but for age estimation in particular it is particularly easy to imagine invasive applications. It could be helpful for the authors to mention why they might want to do age estimation. For example, they even mention that the reason age estimation of young people is hard is do to “ethical limitations”, so why would we want to be able to estimate the age of young people?

**Claims And Evidence:**

Yes

**Requested Changes:**

Requested Changes/Clarifications:

Method and Theory:
1. In what sense is Figure 1 showing “theoretical error bounds”? Aren’t these empirical expectations of errors computed from the datasets? Please clarify.
2. Clarify what is meant by “accurate retrieval of neighbors”. Why is this a hard problem? Isn’t deterministically choosing the nearest neighbors from the training set “accurate”? I think the word “accurate” make this awkward; consider rephrasing.
3. In 3.2.3 and 3.3, clarify which in-context learning method is used and plotted in Figures 2 and 3.
4. Theoretical motivation for the augmented dataset is missing, i.e., compared to just using the inverse density dataset. Is just using the inverse density dataset equivalent to Downsampling in the ablations? If not, it should be one of the ablations, especially since downsampling does quite well in the `few’ category in the ablations. Is the idea of augmented that it trades off in a balanced way between downsampling and vanilla? Figure 6 in the appendix also does not explain why the method does not simply use the Inverse Density dataset alone.
5. It should be clarified what the theory says beyond standard k-nearest neighbors regression theory. E.g., it suggests there is an optimal intermediate k, but does it suggest anything specific beyond that? Is the paper claiming that this is a novel theoretical perspective that also applies to standard k-nearest neighbors?

Experiments:
1. It would be valuable to also compare to RF and linear for a sense of where the method fails compared to standard off-the-shelf approaches.
2. The paper says three “different random seeds, between 0 and 3” were used. Which three?
3. Under “Preprocessing of inputs…” Should "$\tilde{k}_s = \tilde{k}_s =$" be "$k’_s = \tilde{k}_s =$"?
4. What is “VANILLA” in Tables 1 and 2 and Figure 4? This should be clarified before the Ablations section. It looks like an acronym in all caps.
5. Clarify and/or hypothesize why PFN not as strong as Gradient Boosting in the Tabular datasets.
6. Are the neighbors sampled according to a distribution or selected deterministically given the augmented dataset?

Minor changes:

- Fix quote directions, e.g., ‘Prior-data Fitted Networks’ -> \`Prior-data Fitted Networks’, ‘closest’ -> `closest’, etc. “Prior Fitted Networks,” -> ``Prior Fitted Networks’,’ …
- “…covering two facial ages, and one text similarity estimation, as well as…” -> “covering two facial age estimation tasks, one text similarity task, and…”
- Fix citations: e.g., missing parentheses around many citations (probably \citet v. \cite or \citep), and “the authors in Wang & Wang (2024) leveraged” -> “Wang & Wang (2024) leveraged”, “authors in Garg et al.”, … i.e., can generally remove “authors in”
- “…each chunk To ensure…” -> “…each chunk. To ensure...”
- “…expect for the…” -> “…except for the…”
- “et al. (2024)” overlapping other text in Table 1.
- Clarify what bold and underline indicate in tables.
- Figure 4 should be labeled as a Table.

**Strengths And Weaknesses:**

Strengths:
- The skewed distributions of quality in engineering design is a great motivating example. This is an important issue in automated design of anything of especially high quality.
- The general approach of using only the nearest training samples to build the context prompt is simple, intuitive, and practical. I expect it will be used by many practitioners. I’d even guess it is being used by some using LLMs for regression, but without the motivation and theoretical context this paper develops.
- Figure 1 is clear and compelling; a great way to immediately see the problem.
- The empirical results are convincing.

Weaknesses:
- Value of theory is not clear
- Motivation for augmented sampling is not quite clear
- Some experimental details are missing
- Some thought-provoking experimental behavior is not explained

---

> ### Author Response · Authors · 2024-09-06
>
> # Response to Reviewer
>
> Dear Reviewer,
>
> Thank you for your insightful comments. We appreciate your feedback and have addressed each point as follows:
>
> ## Methods and Theory
>
> 1. We apologize for the confusion. We have updated the caption of Figure 1 to refer to "empirical expectations of errors" rather than "theoretical error bounds" to more accurately reflect the nature of the data presented.
>
> 2. We have revised this section to clarify our meaning. What we intended to convey is the challenge of selecting neighbors that are truly similar in terms of output labels, rather than just close in feature space. This is particularly challenging for imbalanced datasets. As highlighted by [1], neighbors selected using metrics like cosine similarity in feature space do not necessarily correspond to similar outputs. We've rephrased this section to avoid the term "accurate" and better explain this concept.
>
> 3. We have updated the captions for Figures 2 and 3 to specify that the GPT2 model is used for in-context learning. Thank you for pointing out this omission.
>
> 4. We appreciate your observation about the need for more clarity regarding the augmented dataset approach. We have made several updates both in the text and also in terms of the experiments to address this:
>    - We have added new ablation using only the Inverse versions alone to provide a more comprehensive comparison with Downsampling.
>    - **The motivation for the augmented approach is to balance the representation of samples from different regions of the label distribution. While methods like Inverse sampling or Downsampling can improve performance in few-shot regions, they may sacrifice performance in majority regions. Our augmented approach aims to maintain good trade-off in performance across all regions.**
>    - We added an explanation on the figure based on the different strategies, using the vanilla distribution will skew the choice of point to the majority region while being accurate in the majority region. However, in the other case, 'inverse sampling' the distribution is now more representative for the minority region, while being still good and accurate with
>
> 5. For in-context learning, both relevance (choosing in-context examples similar to the test input) and diversity among examples are crucial [2]. Our work focuses primarily on relevance and studies its impact across different data regions. This naturally leads us to build upon the intuitive theory of k-NN regression.
>
>    It's important to clarify the similarities and differences between our approach and k-NN. Both methods don't require explicit training and rely on local information for predictions. However, while k-NN uses predefined distance metrics, our in-context learning approach leverages learned distance metrics through meta-training of transformer models.
>
>    Our findings align with current debates in NLP regarding long context versus Retrieval-Augmented Generation (RAG). A paper submitted on arXiv yesterday [3] makes a very similar observation to ours, demonstrating a U-shape curve for performance metrics when adding long context in various NLP tasks, that further substantiates our observations for tabular data.
>
>    Our theoretical framework, while related to k-NN, addresses a different setup. It provides insights in several key aspects:
>
>    1. We analyze the general convergence of the model and guarantee convergence in balanced regression cases. We would like to clarify that for data-scarce regions, comparing the model's expected error to k-NN becomes the unique choice, as both methods don't require training but differ in distance metrics.
>
>    2. **Our work demonstrates how optimal context size varies across different regions of an imbalanced dataset, particularly in few-shot regions. Empirically, we show that in data-scarce regions, adding more samples doesn't necessarily improve in-context learning performance. This is an important finding with significant implications both within engineering and broadly in tabular learning, as it challenges the common assumption that more data always leads to better performance. Our work reveals that in imbalanced scenarios, the relationship between data quantity and model performance is more nuanced and depends on the specific characteristics of the data distribution.**
>
>    3. While theoretical insights for in-context learning are model-specific, we demonstrate empirically that error trends often follow a U-shape curve for performance metrics as the context size increases. This is consistent with k-NN but reveals crucial differences in handling imbalanced data and long contexts in few-shot regions.

---

> > ### Author Response · Authors · 2024-09-06
> >
> > ## Experiments
> >
> > 1. We appreciate your suggestion to include comparisons with Random Forest and linear regression. We have run additional experiments to include these methods. We observe that Random Forest performs competitively, particularly in the 'Many' regions, while our method still maintains an advantage in the 'medium' and 'Few' shot regions.
> >
> > 2. We apologize for the confusion in our original statement. To clarify, we ran the experiments with three specific seeds: 0, 1, and 2. The results reported in the paper are the average across these three experiments.
> >
> > 3. You are correct, it should be k' = \tilde{k}. We have corrected this in the manuscript.
> >
> > 4. 'Vanilla' in our experiments refers to training a ResNet-50 model using standard MSE loss. We updated the manuscript, and changed to MSE, and added the description of the model in the caption.
> >
> > 5. The results obtained are consistent with benchmark methods for tabular data [4,5]. Boosting strategies, perform well in many scenarios, especially in data-rich regions. PFN, while competitive, may not outperform boosting methods consistently due to its different learning paradigm. **Our focus was on exploring in-context learning for imbalanced regression, showing its potential in data-scarce regions rather than outperforming all existing methods across all scenarios.** The performance of PFN could potentially be improved with further optimization of context selection, but this was beyond the current study's scope.
> >
> > 6. We updated the manuscript and specified that neighbor selection is performed deterministically using the sklearn KNN implementation with cosine similarity as distance metric.
> >
> > ## References
> >
> > [1] Steck, H., Ekanadham, C., & Kallus, N. (2024, May). Is cosine-similarity of embeddings really about similarity?. In *Companion Proceedings of the ACM on Web Conference 2024* (pp. 887-890).
> >
> > [2] Ye, J., Wu, Z., Feng, J., Yu, T., & Kong, L. (2023, July). Compositional exemplars for in-context learning. In International Conference on Machine Learning (pp. 39818-39833). PMLR.
> >
> > [3] Yu, T., Xu, A., & Akkiraju, R. (2024). In Defense of RAG in the Era of Long-Context Language Models. arXiv preprint arXiv:2409.01666
> >
> > [4] Rubachev, I., Kartashev, N., Gorishniy, Y., & Babenko, A. (2024). TabReD: A Benchmark of Tabular Machine Learning in-the-Wild. arXiv preprint arXiv:2406.19380.
> >
> > [5] McElfresh, D., Khandagale, S., Valverde, J., Prasad C, V., Ramakrishnan, G., Goldblum, M., & White, C. (2024). When do neural nets outperform boosted trees on tabular data?. Advances in Neural Information Processing Systems, 36.

---

> > > ### Comment · Reviewer_cCTG · 2024-09-11
> > >
> > > Thanks for the updates to the paper and new experiments. You said you've revised the writing to avoid the term "accurate" for discussing retrieval of neighbors. The term still appears to be in the current uploaded version. Do you have a revised version you can upload? Thanks!

---

> > > > ### Author Response · Authors · 2024-09-12
> > > >
> > > > Dear Reviewer,
> > > >
> > > > Thank you for pointing this out. Initially, we revised only the caption of Figure 1, but we’ve now updated the entire document to replace all instances of "accurate" when referring to the retrieval of neighbors. All changes are highlighted in blue for new text and red for old text. The updated version has been uploaded.
> > > >
> > > > We appreciate your feedback!

---

> > ### Comment · Reviewer_cCTG · 2024-09-11
> > **Follow-up on relation to k-NN**
> >
> > W.r.t. point (2) on the theoretical framework that's bolded above, doesn't the same conclusion apply to k-NN? If so, is this a novel insight for k-NN?
> >
> > Could you say k-NN is a special case of your method, where the distance metric is not learned, and the prediction function is the mean over in-context labels?

---

> > > ### Author Response · Authors · 2024-09-12
> > >
> > > Dear Reviewer,
> > >
> > > Thank you for your insightful question. You're correct that similar conclusions would be also applicable to k-NN. One way to view our method is to see it as an extension of k-NN principles, with learned distance metrics via transformer models.
> > > While this could potentially offer novel insights for k-NN, as it wasn't our primary focus, we didn't want to over-claim without a rigorous k-NN-focused study. Our contribution lies in extending these concepts to in-context learning and providing a comprehensive analysis across different regions of imbalanced datasets. We're confident in our insights' applicability to in-context learning. Our visual presentation of error across different regions offers intuitive understanding in this context. A potential future direction could be to rigorously evaluate the expansion of these insights for KNN too.
> > >
> > > We appreciate your question, as it helps clarify the scope and implications of our work.

---

> > > > ### Comment · Reviewer_cCTG · 2024-09-12
> > > >
> > > > Thanks for the response. I agree the insights make sense especially from your empirical visualizations. Since the theory of section 3.2.2 is directly based on the definition of k-NN, i.e., "Under this condition, we can assume that, in the worst case, the model will simply average the labels of the context samples.", the bias-variance decomposition applies directly to k-NN, and it would at least be helpful to say somewhere in that section that there is no or you are not aware of any existing such theory on imbalanced regression w/ k-NN.

---

> > > > > ### Comment · Reviewer_cCTG · 2024-09-12
> > > > >
> > > > > Thanks for the paper updates, it is getting clearer. Maybe instead of "that are representative of this region" more clear could be something like "that are most useful for prediction in this region". Again, w/ "representative of this region" it is clearer than "accurate" but still makes the reader wonder why picking the nearest points does not yield the most "representative".

---

> > > > > > ### Author Response · Authors · 2024-09-19
> > > > > >
> > > > > > Dear Reviewer,
> > > > > >
> > > > > > We sincerely thank you for your thorough engagement with our paper. Your suggestions have been invaluable in improving the clarity of our work. We've revised the text as follows:
> > > > > >
> > > > > > 'This underscores the challenge of retrieving neighboring samples that are most useful for prediction in this region, even when using a small number of context examples. As highlighted by Steck et al. (2024), samples that are close in feature space according to a given distance metric (e.g., cosine similarity) do not necessarily have similar target values. In imbalanced datasets, this discrepancy between proximity in feature space and similarity in target values makes it difficult to select representative samples for the region, as the difference in density makes it more likely to retrieve samples from the majority region.'
> > > > > >
> > > > > > Does this revision address your concerns? We're open to any further suggestions you might have to improve the clarity of this section.

---

> > > > > > > ### Comment · Reviewer_cCTG · 2024-09-19
> > > > > > >
> > > > > > > Thanks for the updates. Including these will significantly improve the paper clarity. The new proposed text makes the claim unambiguous. I recommend the authors do a final minor edit for clarity to make sure this is unambiguous throughout the paper.

---

> > > > > ### Author Response · Authors · 2024-09-19
> > > > >
> > > > > Dear Reviewer,
> > > > >
> > > > > We appreciate your suggestion. We've conducted a literature review and, to the best of our knowledge, we're currently not aware of existing papers specifically addressing this issue. If the paper is accepted, when submitting the camera-ready version of our paper, we'll add a statement in section 3.2.2 to clarify this point (to either cite references we found by that time or state that we are not aware of existing references). Thank you for bringing this to our attention.

---

### Comment · Reviewer_v6aV · 2024-08-17
**Not my area**

This is not in my area of expertise:

Bandits, online learning, reinforcement learning. I cannot possibly give a review that is better than an informed guess.

---

### Author Response · Authors · 2024-09-06

# General Response to Reviewers

Dear Reviewers and AC,

We sincerely thank you for your time and detailed reviews. We appreciate the thoughtful comments and have tried to address them to the best of our ability.

We thank the reviewers for their positive feedback. All reviewers agreed on the following points:

- The paper addresses an important and practical problem of imbalanced regression.
- The proposed approach is innovative and brings new insights to the field.
- The empirical results are convincing and demonstrate the effectiveness of the method.

Additionally:

- Reviewer cCTG found that the motivation example of skewed distributions in engineering design is compelling, and noted that the approach of using nearest training samples for context is simple, intuitive, and practical.
- Reviewer wjE2 appreciated the theoretical error bound analysis and its insights.
- Reviewer nEPj commented that the paper is well-written and easy to follow.

## Changes Made to the Document

We have made the following changes to the document:

1. As requested by reviewer nEPj, we have included a comparison of In-Context Learning (ICL) vs In-Weight Learning (IWL) in a fine-tuning scenario. This provides additional insights into the performance of our method.

2. We have added missing references and included implementation details of baselines as suggested by reviewer wjE2.

3. We have rerun our method on the IMDB dataset using k' = 5 instead of k̃ = 5. The old results are shown in red and the new results in blue for easy comparison.

4. We have included an analysis of the inference time trade-off between ICL and IWL, addressing reviewer wjE2's comment.

5. We have corrected typos, added classifications, and included experiments with Random Forest and Linear Regression models, as suggested by reviewer cCTG.

6. All changes made to the document are highlighted in blue for easy identification by the reviewers.

We believe these changes have significantly improved the paper and addressed the main concerns raised by the reviewers.

---

### Decision · Action_Editor_aU91 · 2024-10-04

**Recommendation:** Accept with minor revision

**Comment:**

The reviewers are in agreement that the paper should be accepted and will be a valuable contribution.

The updates to the paper so far are much appreciated and significantly improve the clarity of the work. There are still some minor typos that should be addressed in a minor revision:

- Citations must be updated to include parentheses when appropriate, as noted by one of the Reviewers
- “retrieving semantically relevant as context” -> “retrieving semantically relevant examples as context”
- “To the best of our knowledge, we are not aware of existing references that specifically address this issue in the context of imbalanced regression with k-NN” -> “To the best of our knowledge, there is no existing work that specifically addresses this issue in the context of imbalanced regression with k-NN”
- ‘Inverse’ -> `Inverse’
- ‘inverse sampling’ -> `inverse sampling’
- “Gradient Boosting,Linear Regression” -> “Gradient Boosting, Linear Regression”

Please upload this revision when you have the chance.

Thanks!
---AE

**Audience:**

Yes

**Claims And Evidence:**

Yes